# Tissue-Specific Effects of the DNA Helicase FANCJ/BRIP1/BACH1 on Repeat Expansion in a Mouse Model of the Fragile X-Related Disorders

**DOI:** 10.3390/ijms26062655

**Published:** 2025-03-15

**Authors:** Diego Antonio Jimenez, Alexandra Walker, Karen Usdin, Xiaonan Zhao

**Affiliations:** Section on Gene Structure and Disease, Laboratory of Cell and Molecular Biology, National Institute of Diabetes and Digestive and Kidney Diseases, National Institutes of Health, Bethesda, MD 20892, USA; diego.jimenez@pennmedicine.upenn.edu (D.A.J.); alexandra.walker@nih.gov (A.W.)

**Keywords:** *FMR1* gene, somatic instability, DNA helicase, FANCJ, repeat expansion diseases, genetic modifier

## Abstract

Fragile X-related disorders (FXDs) are caused by the expansion of a CGG repeat tract in the 5’-UTR of the *FMR1* gene. The expansion mechanism is likely shared with the 45+ other human diseases resulting from repeat expansion, a process that has been shown to require key mismatch repair (MMR) factors. FANCJ, a DNA helicase involved in unwinding unusual DNA secondary structures, has been implicated in a number of DNA repair processes including MMR. To test the role of FANCJ in repeat expansion, we crossed *FancJ*-null mice to an FXD mouse model. We found that loss of FANCJ resulted in a trend towards more extensive expansion that was significant for the small intestine and male germline. This finding has interesting implications for the expansion mechanism and raises the possibility that other DNA helicases may be important modifiers of expansion risk in certain cell types.

## 1. Introduction

Fragile X-related disorders (FXDs) are a group of human genetic disorders caused by an increase in the size of a CGG repeat tract in the 5’-untranslated region of the fragile X messenger ribonucleoprotein 1 (*FMR1*) gene on the X chromosome [1,2]. Typical alleles have about 30 repeats [3]. Full mutation (FM) alleles have >200 CGG repeats [4,5] that result in the absence of the *FMR1* gene product, FMRP, by causing repeat-mediated silencing [6]. The loss of FMRP results in fragile X syndrome (FXS), the most common cause of inherited intellectual disability [2]. FM alleles arise from expansion of a premutation (PM) allele with 55–200 repeats that occurs through maternal transmission [7]. In addition to the risk of having a child with FXS, carriers of PM alleles are at risk of having one or more of fragile X premutation conditions (FXPACs), including fragile X-associated tremor/ataxia syndrome (FXTAS), fragile X-associated primary ovarian insufficiency (FXPOI), and fragile X-associated neuropsychiatric disorder (FXAND) [8]. In addition to the propensity of the PM allele to expand in the germline [9], expansion also occurs in somatic cells with expansion risk increasing as repeat number becomes greater [10].

Repeat expansion is also responsible for more than 45 other human genetic disorders, referred to collectively as the Repeat Expansion Diseases (REDs) [11]. The REDs include Huntington’s disease (HD), myotonic dystrophy type 1 and 2 (DM1 and DM2), and many spinocerebellar ataxias (SCAs). The age of onset and severity of many of these diseases are related to the extent of somatic expansion [12,13,14,15,16]. Since many of these diseases lack effective treatments, reducing somatic expansion may be a valuable treatment approach [17,18]. However, although many genetic modifiers associated with somatic expansion have been identified [19], the mechanism of repeat expansion is not fully understood. Therefore, there may be important genetic modifiers of expansion risk that remain to be identified that could be useful therapeutic targets.

We have previously identified a number of DNA repair genes associated with repeat expansion in an FXD mouse model [20], including those encoding proteins involved in mismatch repair (MMR) [21,22,23,24,25,26], as well as those from other DNA repair pathways [27]. Many of these same factors have been implicated in animal and cell models of other REDs [20,28]. Furthermore, genome-wide association studies (GWASs) and the analysis of single-nucleotide polymorphisms have shown that some of these genes are also modifiers of somatic instability in diseases like HD [13,29], DM1 [30], SCAs [31], and the FXDs [32]. Thus, the REDs are all likely to share a common expansion mechanism that is recapitulated in our FXD mouse model.

Notably, while MMR deficiency has been implicated in the microsatellite instability (MSI) associated with certain cancer predisposition syndromes like Lynch syndrome, certain MMR proteins are actually required for repeat expansion in the REDs [33]. Thus, the expansion mechanism responsible for the FXDs and other REDs may be quite different from classical MSI. The requirement for MutSβ, an MMR complex, which binds small insertions or deletions [34,35,36], suggests that the substrate for expansion contains one or more loop-outs. Since transcription is also required for expansion [37,38], it may be that these loop-outs form as a result of misalignment of the repeat region behind the transcription complex. Many of the disease-associated repeats form non-B DNA structures, such as hairpins, G-quadruplexes (G4-DNA), H-DNA, or cruciforms, are thought to play a role in expansion perhaps by favoring such strand misalignment [39].

FANCJ, also known as BRCA1 Interacting Protein 1 (BRIP1) or BRCA1-associated C-terminal DNA helicase (BACH1) [40], is a DEAH (Asp-Glu-Ala-His) superfamily 2 helicase that has been hypothesized to play a role in repeat expansion [41]. In vitro, it acts in a 5’ to 3’ direction to unwind a variety of DNA substrates including 5′ flaps, D-loops [42], 5′ tailed triplexes [43], and G4-DNA structures [44,45]. While it is best known for its role in resolving interstrand crosslinks (ICLs) as part of the Fanconi anemia pathway (FA) [46,47,48], it has been implicated in other forms of double strand break repair (DSBR) as well as MMR [40,41,49,50]. FANCJ interacts with FAN1 [51], a nuclease that we have shown to protect against expansion independently of the FA pathway [52,53]. FANCJ also interacts with MLH1, a protein required for repeat expansion [25,54,55], in both the repair of ICLs [56] and in MMR [57,58,59]. In addition, it has been implicated in the resolution of G4/R-loops at sites of transcription–replication conflict [60].

Thus, FANCJ has many features that make it a good candidate modifier of repeat expansion. To test the role of FANCJ in repeat expansion, we crossed *FancJ* mutant mice to an FXD mouse model and compared the effect of loss of FANCJ in different tissues. We found that loss of FANCJ increased expansion in some, but not all organs. This makes FANCJ the first helicase shown to be a modifier of repeat expansion in a mouse REDs model. This finding has implications for the expansion mechanism and raises the possibility that other helicases may be important modifiers of expansion risk in certain cell types including neurons.

## 2. Results

### 2.1. Loss of FANCJ Increases Somatic Expansion in the Small Intestine and Male Germline

To examine the role of FANCJ in repeat expansion in an FXD mouse model, we crossed FXD mice to mice with a null mutation in *FancJ*. We then compared the extent of instability in different tissues of 7 *FancJ^+/+^* and 8 *FancJ^−/−^* 6-month-old male mice matched for repeat size. As can be seen in Figure 1, loss of FANCJ resulted in an approximate 31% increase in the extent of expansion in the small intestine, one of the most expansion-prone tissues. Testes, which are somewhat less prone to expansion, showed a small but consistent increase in repeat number that amounted to an approximate 19% increase relative to WT animals. The kidney, tail, and cerebellum also showed significant increases in the number of repeats added. However, the significance in these tissues did not survive correction for multiple testing. The remaining tissues showed a trend towards expansion that was not significant even before correction for multiple testing.

The analysis of repeat expansion in females is complicated by the fact that the *Fmr1* gene is X-linked. Thus, because of X inactivation, only a fraction of the expansion-prone alleles is on the active X chromosome and thus able to expand [37]. This results in a repeat PCR profile containing a mixture of expanded and non-expanded alleles (Appendix A). The PCR profiles can be complex to interpret if the repeat number in the two allele populations overlap. To avoid this issue, we limited our analysis of the effect of the loss of FANCJ to those ovary samples that produced a unimodal PCR profile consistent with most of the expansion-prone alleles being on the active X chromosome. No significant difference was seen between the number of repeats added to the FXD allele in *FancJ^+/+^* and *FancJ^−/−^* mice (Appendix A).

To specifically assess the role of FANCJ in the male germline, we collected sperm from 6-month-old animals matched for repeat numbers as described above. As can be seen from Figure 2B, the number of repeats added in the sperm from the *FancJ^−/−^* animals was significantly higher than that of *FancJ^+/+^* animals. Small-pool (SP)-PCR on sperm samples showed both a significant increase in the number of sperm containing an expanded allele as well as significant decrease in the number of sperm carrying alleles that had contracted (Figure 2C).

To assess the effect of FANCJ on maternally transmitted expansions, we compared the repeat number in the offspring of breeding pairs in which the mother carried the expanded allele and both parents were either *FancJ^+/+^* or *FancJ^−/−^*. No difference in the repeat size distribution of the progeny was seen (Figure 2D), suggesting that FANCJ has no effect on the frequency or size of maternally transmitted expansions.

A similar study of the effect of FANCJ in the organs of 12-month-old mice showed a significant increase in expansion in the small intestine and cortex (Appendix A). However, neither of these differences survived correction for multiple testing, suggesting that FANCJ’s importance in preventing expansion decreases with age in the small intestine.

### 2.2. Loss of FANCJ Has No Effect on Somatic Contractions in the Small Intestine

Although the disease-associated repeats show an expansion bias, contractions are also found in cell and mouse models of different REDs [22,25,61,62,63,64,65]. These contractions could contribute to the mosaicism seen in some FXD patients that may modulate the disease risk. To test if FANCJ loss affected the extent of somatic contraction, we performed a small-pool PCR (SP-PCR) on the small intestine samples. The single allele resolution of the SP-PCR allows us to detect stochastic contractions that can be masked by bulk PCR. As can be seen in Figure 3, both the distribution of total alleles and expanded alleles are significantly different in *FancJ^+/+^* and *FancJ^−/−^* animals. The shift in the distribution of alleles in *FancJ^−/−^* mice is consistent with the increased expansion seen in the bulk PCR. However, there was no significant difference in the number or distribution of contracted alleles, suggesting that FANCJ does not modulate the frequency or extent of somatic repeat contractions.

### 2.3. Loss of FANCJ Does Not Affect Fmr1 Transcription or the Levels of Key MMR Proteins Involved in Expansion

We previously found that transcription or an open chromatin state is required for expansion in the FXD mouse model [37] and evidence from a variety of models suggests that transcription is a driver of expansions [38,66,67,68]. FANCJ has the potential to affect transcription by resolving secondary structures formed on the template [69] and transcription–replication conflicts [60]. We thus tested the effect of the loss of FANCJ on *Fmr1* expression by qPCR to assess whether this might account for FANCJ’s effect in the small intestine and testes. However, although there were variations in *Fmr1* mRNA levels in the small intestine of animals with the same genotype, we found no significant difference in *Fmr1* mRNA levels between *FancJ^+/+^* and *FancJ^−/−^* animals in either the small intestine or the testes (Figure 4).

Mice homozygous for a large disruption of *FancJ* (*Fancj^GT/GT^*) have been shown to have elevated expression of MLH1 [70]. Since the levels of MutLα and MutLγ, which are important modifiers of expansion [24,25,26,54,55,71,72,73,74], are dependent on the levels of MLH1, this could potentially contribute to the increased level of expansion we observed. However, Western blotting of protein extracts from the small intestine and testes showed no significant effect on the MLH1 levels (Figure 5). The differences between the findings of the two studies may be related to the differences in the *FancJ* alleles (Gene trap disruption in the case of the *Fancj^GT/GT^* mice vs. CRISPR knockout in the case of the *FancJ*^−/−^ strain we used) or the strain differences in the mouse models (129P2/OlaHsd background and a C57BL/6J and 6N mixed background, respectively). Whatever the explanation, since we saw no difference between WT and null mice in the levels of MLH1, this does not explain the increased levels of expansion observed in the absence of FANCJ.

## 3. Discussion

Our analysis of repeat expansion in *FancJ* null FXD mice revealed a significant role for the FANCJ helicase in protecting against expansion in some organs, at least in younger animals. However, only the significance in the small intestine and testes survived correction for multiple testing (*p* < 0.05). Sperm samples mirrored the effect seen in whole testes. Thus, FANCJ also has a protective effect in both a subset of somatic cells as well as the male germline. No effect of the loss of FANCJ was seen in the liver of 6-month-old FXD mice, consistent with what had been observed from CRISPR-mediated reduction of FANCJ/BRIP1 in the liver of an HD mouse model [74]. Furthermore, while more expansion was seen in the small intestine and cortex of 12-month-old *FancJ*^−/−^ animals, neither effect survived correction for multiple testing, suggesting that FANCJ’s protective effect declines with age.

We have previously shown that loss of FANCD2, which is essential for the FA pathway of DNA repair, has no effect on expansion in any of the mouse organs tested including the small intestine and testes [53]. Thus, FANCJ is acting to protect against expansion independently of the FA pathway. FANCJ is a helicase implicated in the removal of DNA secondary structures [43,44,45]. Since disease-associated repeats form similar structures that are thought to play an important role in the expansion process [39], it may be that FANCJ protects against repeat expansion by facilitating their removal. While GWAS to date have not implicated FANCJ as a genetic modifier of human expansion risk, FAN1, a protein partner of FANCJ in a variety of repair pathways [75], is one of the most important known modifiers of somatic expansion in REDs patients [13,29,32,76]. Both FAN1 and FANCJ interact with MLH1 [51,56,77], and both are associated with microsatellite instability [41,50,78]. It may be that the two proteins act together to protect against repeat expansion. However, since FAN1’s effect is seen in a broader range of tissues, if indeed FANCJ and FAN1 cooperate in protecting against expansion, other factors must be able to compensate for FANCJ’s loss.

The absence of a significant protective effect of FANCJ in organs other than the small intestine and testes does not necessarily mean that helicases are not important in those tissues. Mammalian cells have many helicases that could compensate for the absence of FANCJ [79], and it may be that the importance of any particular helicase depends on its levels relative to the other helicases. In this context, it should be noted that the RTEL1 helicase has been shown to protect against large expansions in a plasmid-based model of CAG expansions in the SVG-A astrocytic cell line [80], while DDX11, REQ1, and WRN helicases do so in a plasmid-based model of large GAA expansions in HEK293-derived cells [81]. However, at least in the case of the GAA model system, the expansions detected were not MMR-dependent [81]. Thus, whether these helicases protect against the MMR-related somatic expansions observed in patients or the somatic expansions detected in the FXD mouse model remains to be tested. While we do not have reliable antibodies to quantify the relative amount of FANCJ in different cell types, transcriptome data from the Mouse ENCODE project [82] show higher levels of the *FancJ* transcript in the testes and small intestine than in some of the other organs tested. Although transcript levels do not always correlate with protein levels, it may be that FANCJ is particularly abundant in those organs and thus its loss would be more noticeable. Additionally, since the small intestine and testes are among the most expansion-prone tissues we have examined, they likely have high levels of the substrates that cause repeat expansion. Under these circumstances the effect of the loss of a single helicase like FANCJ may be more readily apparent.

The SP-PCR data in sperm and the small intestine suggest that FANCJ does not influence the contraction frequency. Thus, the apparent decrease in FANCJ’s effect with age is unlikely to be due to a direct effect of FANCJ on contraction. Rather, it may be related to the age-related changes in the cellular environment, including perhaps the increased expression of other helicases, the reduced level of the expansion substrates by alterations in transcriptional activity, chromatin state, or DNA repair efficiency.

Since FXS is congenital and the transcriptionally inactive allele does not expand somatically, and the symptoms of FXPAC do not involve either the small intestine or the testes, FANCJ’s contribution to disease pathology may be limited to its influence on the size of the paternally transmitted allele. However, the fact FANCJ protects against expansion in some somatic cells as well as the male germline not only establishes this helicase as a potential modifier of expansion risk in these cells but also raises the possibility that other helicases are modifiers of expansion risk in other cell types in FXDs as well as other REDs.

## 4. Materials and Methods

### 4.1. Reagents and Services

Reagents and primers were from Thermo Fisher Scientific (Waltham, MA, USA) unless otherwise stated. Capillary electrophoresis of fluorescently labeled PCR products was carried out by the Roy J Carver Biotechnology Center, University of Illinois (Urbana, IL, USA) and Psomagen (Rockville, MD, USA).

### 4.2. Mouse Generation, Breeding, and Maintenance

Sperm from *FancJ* mutant mice were obtained from the Jackson Laboratory (Bar Harbor, ME, USA; JAX stock #042134-JAX; C57BL/6NJ-*Brip1^em1(IMPC)J^*/Mmjax), and mice were recovered by the NIDDK Laboratory Animal Sciences section (LASS) using standard procedures. The FXD mice have been previously described [83]. *FancJ* heterozygous FXD mice were generated by crossing *FancJ* mutant mice with FXD mice. These heterozygous mice were then crossed again to generate FXD mice homozygous for the *FancJ* mutation or *FancJ* WT. Since the *FancJ* mutant mice were on C57BL/6NJ background, and the FXD mice were on C57BL/6J background, all mice used in this study were on a C57BL/6J and 6N mixed background. Mice were maintained in a manner consistent with the Guide for the Care and Use of Laboratory Animals (NIH publications no. 85-23, revised 1996) and in accordance with the guidelines of the NIDDK Animal Care and Use Committee, who approved this research (ASP-K021-LMCB-21).

### 4.3. DNA Isolation

DNA for genotyping was prepared from mouse tail samples collected at 3 weeks of age or at weaning using the KAPA Mouse Genotyping Kit (Roche Diagnostics, Indianapolis, IN, USA). Mice aged 6 or 12 months were euthanized using compressed CO_2_ followed by cervical dislocation. A variety of tissues were collected from these mice, and DNA was isolated using a Maxwell^®^ 16 Mouse Tail DNA Purification kit (Promega, Madison, WI, USA) according to the manufacturer’s instructions. A 5 cm section of the jejunum was collected as the small intestine sample as previously described [84]. Sperm collection and DNA preparation were as previously described [85].

### 4.4. Genotyping and Analysis of Repeat Number

*FancJ* genotyping was performed using the KAPA mouse genotyping kit (Roche Diagnostics, Indianapolis, IN, USA) according to manufacturer’s instructions with primers JAX-32103 (5′-TTGACATGTTTATAAAAC CAGCAGA-3′) and JAX-32106 (5′-AACTAGCAGTCTGAAATATCAAGAAGT-3′) to detect the 542-bp WT *FancJ* allele or the 133-bp mutant *FancJ* allele. The PCR mix for the *FancJ* allele contained 2 μL template DNA, KAPA2G Fast HotStart Genotyping Mix (Roche Diagnostics, Indianapolis, IN, USA) according to the manufacturer’s instructions, and 0.5 μM each of the primers. The *FancJ* allele PCR conditions were 95 °C for 3 min; 35 cycles of 95 °C for 15 s, 60 °C for 15 s and 72 °C for 15 s; followed by 72 °C for 3 min. Genotyping and repeat size analysis of the *Fmr1* allele was conducted using a fluorescent PCR assay with fluorescein amidite (FAM)-labeled primer pairs FAM-FraxM4 (FAM-5′-CTTGAGGCCCAGCCGCCGTCGGCC-3′) and FraxM5 (5′-CGGGGGGCGTGCGGTAACGGCCCAA-3′) as previously described [26]. The PCR products were resolved by capillary electrophoresis on an ABI Genetic Analyzer, and the resultant FSA files were displayed using a previously described custom R script [86] that is available upon request. Repeat size in tail sample collected from mice at 3 weeks of age or at weaning was used as an indicator of the repeat size of the original inherited allele. The change in repeat numbers was simply determined by subtracting the number of repeats in the modal allele from the number of repeats in the original inherited allele.

### 4.5. Small-Pool PCR

Small-pool PCR analysis used a nested PCR strategy with an input of 12 pg of genomic DNA per PCR, consistent with a single-molecule level reported previously [87]. The first rounds of the PCR used the primers Not_mFrax-C (5′-AGTTCAGCGGCCGCGCTGGGGAGCGTTTCGGTTTCACTTCCGGT-3′) and Not_Frax-R4 (5′- CAAGTCGCGGCCGCCTTGTAGAAAGCGCCATTGGAGCCCCGCA-3′) in a 10 μL PCR mix described previously [85]. One microliter of this PCR product was then used as the template in a second round of the PCR using the FAM-FraxM4 and FraxM5 primers as described above. PCR products were resolved by capillary electrophoresis on an ABI Genetic Analyzer.

### 4.6. Quantitation of mRNA

Total RNA was isolated from mouse tissues using the Maxwell^®^ 16 LEV simplyRNA tissue Kit (Promega, Madison, WI, USA) according to the manufacturer’s instructions. Briefly, mouse tissues were homogenized using the Precellys lysing kits (Bertin Technologies, France), and 200 μL of the homogenate was added to the cartridge and processed on the Maxwell^®^ 16 Instrument (Promega, Madison, WI, USA). The total RNA was reverse transcribed using a SuperScript^®^VILO^TM^ cDNA synthesis kit (Thermo Fisher Scientific, Waltham, MA, USA) following the manufacturer’s protocol. A real-time PCR was performed in triplicate using TaqMan^®^ Fast Advanced Master mix (Thermo Fisher Scientific, Waltham, MA, USA), 2 μL of the cDNA, the Taqman probe-primer pairs (Thermo Fisher Scientific, Waltham, MA, USA) of the *Gapdh* (Mouse GAPDH Endogenous Control, VIC^®^/MGB Probe, primer limited), and the Taqman probe-primer pairs of *Fmr1* (Mm01339582_m1) on a StepOnePlus Real-Time PCR System (Thermo Fisher Scientific, Waltham, MA, USA) following the manufacturer’s protocol. Relative *Fmr1* mRNA levels were normalized to the endogenous *Gapdh* level.

### 4.7. Western Blot

Total protein extracts were prepared from flash frozen tissues collected from 6-month-old mice. Tissue samples were washed with cold PBS supplemented with cOmplete, Mini, EDTA-free protease inhibitor cocktail (Roche Diagnostics, Indianapolis, IN, USA) before homogenized. Small intestine samples were homogenized in a RIPA lysis buffer containing 150 mM NaCl, 50 mM Tris-HCl pH 8.0, 1.0% NP-40, 0.5% deoxycholate, 0.1% sodium dodecyl sulfate (MilliporeSigma, Burlington, MA, USA), and protease inhibitor cocktail (MilliporeSigma, Burlington, MA, USA) according to the manufacturer’s instructions, and testes samples were homogenized in the T-PER tissue protein extraction reagent (Thermo Fisher Scientific, Waltham, MA, USA) with protease inhibitor cocktail (MilliporeSigma, Burlington, MA, USA) according to the manufacturer’s instructions. Tissue samples were homogenized twice for 20 s at 6400 RPM with 30 s pause in a soft tissue homogenizing kit (Bertin Technologies, Berlin, Germany) using the Precellys 24 tissue homogenizer (Bertin Technologies, Berlin, Germany). Lysates were then incubated on ice for 10 min before centrifuged at 13,000 RPM for 10 min at 4 °C. The supernatant was transferred to a new pre-chilled tube, and the protein concentration was determined using the Bio-Rad Protein Assay Dye Reagent Concentrate (Bio-Rad, Hercules, CA, USA). After adding reducing agent (Thermo Fisher Scientific, Waltham, MA, USA)and LDS sample buffer (Thermo Fisher Scientific, Waltham, MA, USA) according to the manufacturer’s instructions, proteins were incubated for 10 min at 70 °C then 30 μg protein of the small intestine or 15 μg protein of testes was resolved by electrophoresis on NuPAGE 3–8% Tris-Acetate gels (Thermo Fisher Scientific, Waltham, MA, USA) and transferred to the nitrocellulose membrane using the Bio-Rad Trans-Blot Turbo Transfer System (Bio-Rad, Hercules, CA, USA) according to the manufacturer’s instructions. Membranes were blocked in a 5% Amersham ECL Prime Blocking Agent (Cytiva, Marlborough, MA, USA) in TBST for one hour at room temperature and then incubated overnight at 4 °C with MLH1 antibodies (ab92312, Abcam, Cambridge, UK) at a dilution of 1:10,000. Membranes were then washed 4 times in TBST and incubated for 1 h at room temperature with the secondary ECL HRP-linked Rabbit IG, HRP-linked antibodies (Cytiva, Marlborough, MA, USA) at a dilution of 1:5000. After washing 4 times in TBST and once with TBS, the blots were incubated with an Amersham ECL Western Blotting detection reagent (Cytiva, Marlborough, MA, USA), and the signals were imaged using a ChemiDoc imaging system (Bio-Rad, Hercules, CA, USA). Beta-actin was used as a loading control. After washing with TBST and incubating for 1 h at room temperature with β-actin antibodies (ab8227, Abcam, Cambridge, United Kingdom) at a dilution of 1:5000, the blot was then processed and imaged as before. The amount of MLH1 relative to that of the β-actin was determined by using the Fiji (ImageJ) 2.16.0 software.

### 4.8. Statistical Analyses

Statistical analyses were performed using GraphPad Prism 10.2. For comparisons of repeat number changes in tissue samples between *FancJ^+/+^* and *FancJ^−/−^* mice, statistical significance was assessed using the unpaired t-test with Holm–Šídák’s multiple comparisons correction. Small-pool PCR data of sperm were analyzed using Fisher’s Exact Test (Social Science Statistics) for a statistical significance between expansions, contractions, and no change in alleles of different genotypes. Maternal transmission data and the small-pool PCR data of the small intestine were analyzed using the Mann–Whitney U test. Differences in mRNA and protein levels between *FancJ^+/+^* and *FancJ^−/−^* mice was assessed using the two-tailed unpaired t-test.

## Figures and Tables

**Figure 1 ijms-26-02655-f001:**
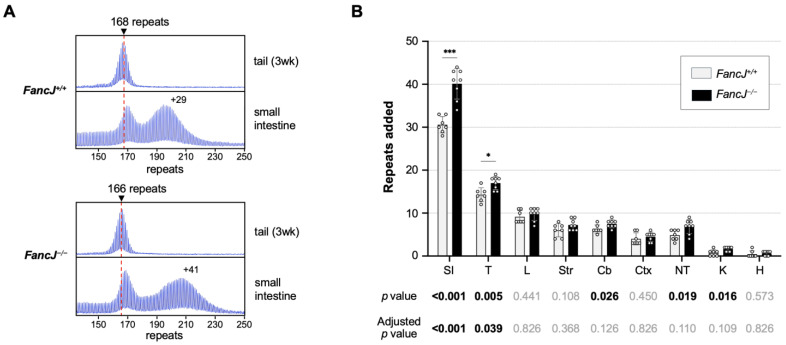
The effect of loss of FANCJ on repeat expansion in different tissues of an FXD mouse model. (**A**) Representative repeat PCR profiles from tail DNA taken at 3 weeks (3 wk) and the small intestine of 6-month-old *FancJ^+/+^* and *FancJ^−/−^* FXD male mice with ~167 repeats. The dashed lines represent the sizes of the original inherited alleles as ascertained from the tail DNA taken at 3 weeks. The number associated with each profile indicates the change in repeat number relative to the original inherited allele. (**B**) Comparison of the repeat added number in the indicated organs of 6-month-old *FancJ^+/+^* and *FancJ^−/−^* FXD mice with an average of 167 repeats in the original allele. The data represent the average of 7 *FancJ^+/+^* and 8 *FancJ^−/−^* mice with 165–170 repeats. The error bars indicate the standard deviations of the mean. Each dot represents one animal. In each organ, the number of repeats added for different genotypes were compared using unpaired t-test with Holm–Šídák’s multiple comparisons correction. The *p*-values and adjusted *p*-values are listed in the table below. ***, *p* < 0.001; *, *p* < 0.05. SI, small intestine; T, testes; L, liver; Str, striatum; Cb, cerebellum; Ctx, cortex; NT, tail; K, kidney; H, heart.

**Figure 2 ijms-26-02655-f002:**
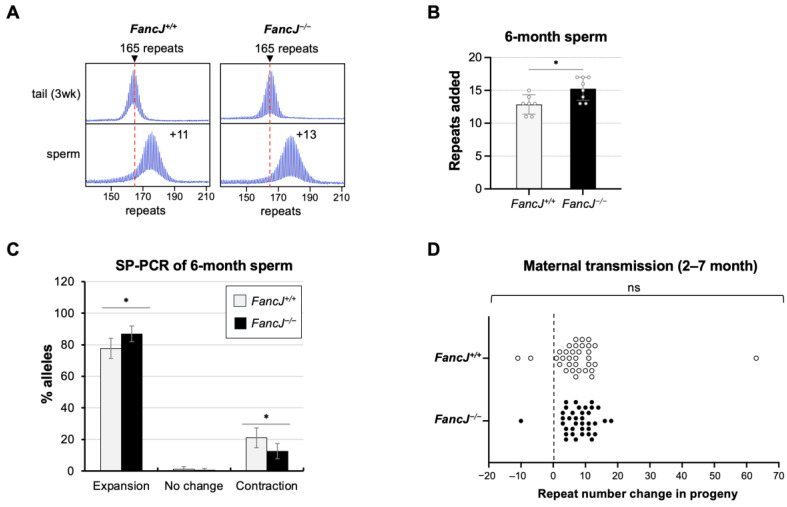
The effect of loss of FANCJ on repeat instability in the gametes and intergenerational transmission. (**A**) Representative repeat PCR profiles of sperm from 6-month-old *FancJ^+/+^* and *FancJ^−/−^* FXD male mice with 165 repeats. The dashed lines represent the sizes of the original inherited alleles as ascertained from the tail DNA taken at 3 weeks. The number associated with each profile indicates the change in repeat number relative to the original inherited allele. (**B**) Comparison of the repeat added number in the sperm of 6-month-old *FancJ^+/+^* and *FancJ^−/−^* FXD mice with an average of 167 repeats in the original allele. The data represent the average of 7 *FancJ^+/+^* and 8 *FancJ^−/−^* mice with 165–170 repeats. The error bars indicate the standard deviations of the mean. Each dot represents one animal. The number of repeats added was compared using the two-tailed unpaired t-test. *, *p* < 0.05. (**C**) Small-pool PCR (SP-PCR) of sperm DNA isolated from 6-month-old *FancJ^+/+^* and *FancJ^−/−^* FXD mice with 168–170 repeats in the original allele. A total of 166 positive PCR products from 2 *FancJ^+/+^* males and 183 positive PCR products from 2 *FancJ^−/−^* males were analyzed. The error bars represent the 95% confidence interval. The difference between the number of expansions, contractions, and unchanged alleles in the two groups of animals was evaluated by Fisher’s exact test. *, *p* < 0.05. (**D**) Distribution of the change in repeat numbers in the progeny of the mothers at the age of 2–6 months old with 157–158 repeats in the original allele. Thirty-three pups from 3 *FancJ^+/+^* and 36 pups from 3 *FancJ^−/−^* females were analyzed. The dashed line indicates alleles of the same size as the mother’s original allele. The distribution of allele sizes in the pups were analyzed by Mann–Whitney U test. ns, not significant.

**Figure 3 ijms-26-02655-f003:**
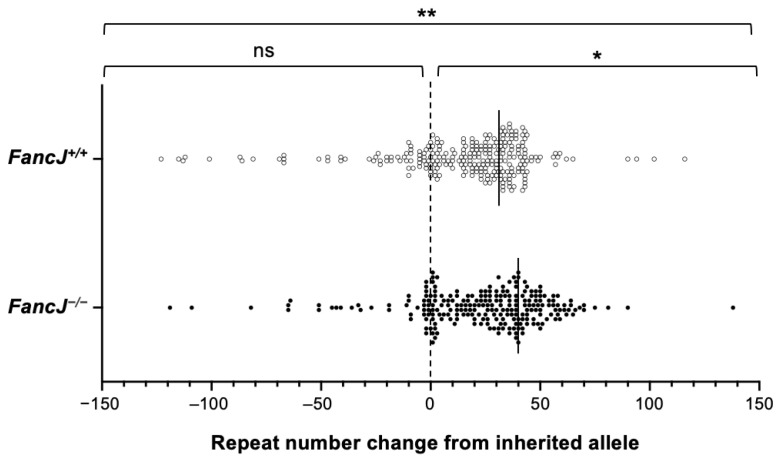
The effect of loss of FANCJ on repeat contractions in the small intestine. The SP-PCR of DNA isolated from the small intestines of 6-month-old *FancJ^+/+^* and *FancJ^−/−^* FXD mice with 168–170 repeats in the original allele. A total of 261 positive PCR products from 3 *FancJ^+/+^* males and 239 positive PCR products from 3 *FancJ^−/−^* males were analyzed. The dashed line indicates alleles of the same size as the original allele. The solid lines indicate the average of the repeat added number measured by bulk PCR. Distribution of the change in repeat number of total, expansion, and contraction alleles were analyzed separately by Mann–Whitney U test. **, *p* < 0.01; *, *p* < 0.05; ns, not significant.

**Figure 4 ijms-26-02655-f004:**
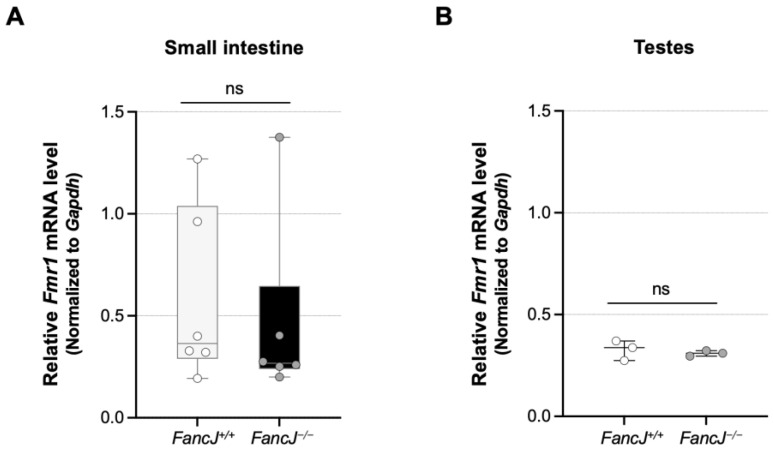
The effect of loss of FANCJ on the transcription level of CGG containing *Fmr1* gene. Box plot of *Fmr1* mRNA levels in the small intestine (**A**) and testes (**B**) in 6-month-old male mice with an average of 167 repeats in the original allele. The amount of *Fmr1* mRNA was evaluated by real-time quantitative PCR as described in Materials and Methods. The *Fmr1* mRNA levels are expressed relative to the levels of a house keeping gene, *Gapdh*. The data of the small intestine are based on 6 *FancJ^+/+^* and 6 *FancJ^−/−^* mice with 158–175 repeats, and the data of testes are based on 3 *FancJ^+/+^* and 3 *FancJ^−/−^* mice with 158–175 repeats. Each dot represents one animal. The box represents the 25–75th percentiles, and the median is indicated. The whiskers show the minimum and maximum values. Significance was assessed using two-tailed unpaired t tests. ns, not significant.

**Figure 5 ijms-26-02655-f005:**
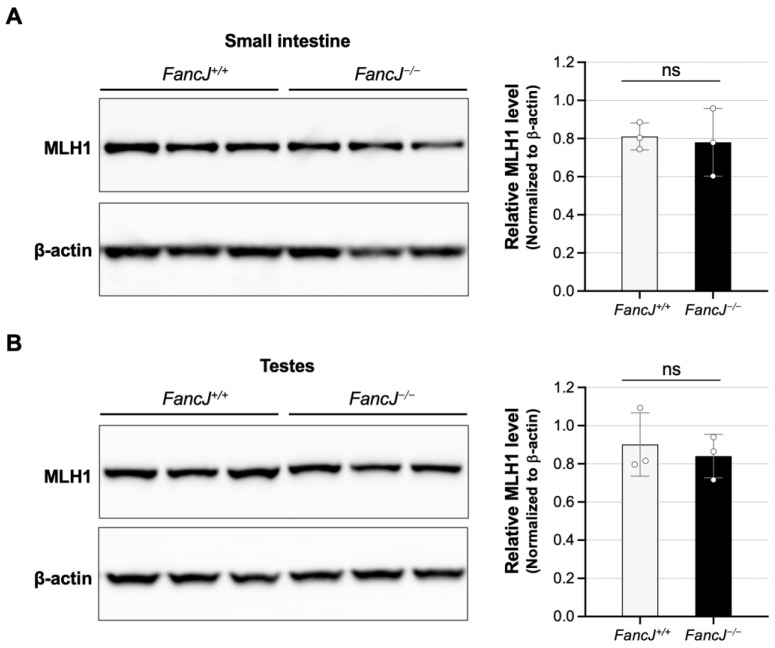
The effect of loss of FANCJ on MLH1 protein levels. The amount of the MLH1 protein in the small intestine (**A**) and testes (**B**) in 6-month-old male mice was evaluated by Western blot as described in Materials and Methods. The protein quantitation was measured by Fiji (ImageJ). The MLH1 protein levels were normalized to a house keeping gene, β-actin. Each dot represents one animal. The error bars indicate the standard deviations of the mean. Significance was assessed using two-tailed unpaired t tests. ns, not significant.

## Data Availability

All data generated or analyzed during this study are included in this published article and its Appendix A files.

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
