# Peer review of "Tissue-Specific Effects of the DNA Helicase FANCJ/BRIP1/BACH1 on Repeat Expansion in a Mouse Model of the Fragile X-Related Disorders"

_ijms, 2025, doi:10.3390/ijms26062655_

Round 1
Reviewer 1 Report
Comments and Suggestions for Authors
The authors demonstrated that FANCJ plays a role in regulating repeat expansion in a FXD mouse model. They also showed that the effects of FANCJ are tissue- and age-specific. The overall findings are interesting, and the manuscript is well-organized. However, several concerns need to be addressed:
- The tissue-specific effect of FANCJ depletion on repeat expansion is intriguing. Why are the small intestine and testis particularly sensitive to the loss of FANCJ? Is it due to lower protein expression of FANCJ in these tissues compared to others?
- Intellectual disability is a significant phenotype of FXD. However, FANCI depletion had no notable influence on the nervous system in FXD mice. Does this imply that FANCI does not play a critical role in the onset of FXD?
- Why does FANCJ depletion influence expansion in the small intestine and testis of 6-month-old mice but not in those of 12-month-old mice? This is quite intriguing, and the authors only state that FANCJ's role in preventing expansion decreases with age. Does this phenomenon result from an increase in repeat expansion in WT mice, a repeat contraction in FANCJ mice, or other potential explanations? The authors should discuss this point further.
- Does FANCJ depletion have any effect on the Fmr1 allele in non-FXD mouse models?
- Are FANCJ mutations found in patients with REDs? Providing such evidence would enhance the clinical significance of the findings in this work.
- Some parts of Figure 1 are duplicated due to typographical errors.
- “uL” should be corrected to “μL”.
Author Response
- The tissue-specific effect of FANCJ depletion on repeat expansion is intriguing. Why are the small intestine and testis particularly sensitive to the loss of FANCJ? Is it due to lower protein expression of FANCJ in these tissues compared to others?
Response 1: Mammalian cells express multiple helicases that could potentially compensate for the absence of FANCJ in different cells. While we do not have reliable antibodies to quantify the relative amount of FANCJ in different cell types, transcriptome data from the Mouse Encode project shows higher levels of FancJ transcript in the testes and small intestine than in some of the other organs tested. Although transcript levels do not always correlate with protein levels, it may be that FANCJ is particularly abundant in those organs and thus its loss would be more noticeable. Additionally, since the small intestine and testes are among the most expansion-prone tissues we’ve examined, they presumably have high levels of the substrates that cause repeat expansion. Under these circumstances the effect of the loss of a single helicase like FANCJ may be more readily apparent. We have added a paragraph in the Discussion to discuss this issue.
- Intellectual disability is a significant phenotype of FXD. However, FANCI depletion had no notable influence on the nervous system in FXD mice. Does this imply that FANCI does not play a critical role in the onset of FXD?
Response 2: FXS, the FXD associated with intellectual disability, is a congenital disorder resulting from repeat-mediated gene silencing in early development. Since the alleles are fully methylated, and transcription is required for expansion, no somatic expansion would be seen. As such somatic instability in the affected individual is not likely to play a role in the onset of FXS, and thus by extension neither would FANCJ. In addition, since the symptoms of the other fragile X premutation-associated conditions (FXPAC), including FXTAS, FXPOI, and FXAND, do not involve either the small intestine or the testes, FANCJ’s contribution to disease pathology may be limited to its effect on the size of the paternally transmitted allele. However, our data implicating a helicase in the protection against expansion, raises the possibility that other helicases do contribute to somatic instability in the FXDs as well as other REDs. We have tried to clarify this in our discussion.
- Why does FANCJ depletion influence expansion in the small intestine and testis of 6-month-old mice but not in those of 12-month-old mice? This is quite intriguing, and the authors only state that FANCJ's role in preventing expansion decreases with age. Does this phenomenon result from an increase in repeat expansion in WT mice, a repeat contraction in FANCJ mice, or other potential explanations? The authors should discuss this point further.
Response 3: Our SP-PCR data in sperm and small intestine suggest that FANCJ does not influence the contraction frequency. Thus, the effect of age is unlikely to be due to a direct effect of FANCJ on contraction. It may be that increased expression of other helicases might compensate for any loss of FANCJ. We have modified the discussion to elaborate on these points.
- Does FANCJ depletion have any effect on the Fmr1 allele in non-FXD mouse models?
Response 4: We have not investigated the effects of FANCJ depletion on the Fmr1 allele in non-FXD mouse models. However, we anticipate that since the effect of FANCJ would likely be more apparent on larger alleles, it may be very difficult to determine any effect on the normal mouse Fmr1 repeat which is only 8 repeats long and shows little, if any, instability.
- Are FANCJ mutations found in patients with REDs? Providing such evidence would enhance the clinical significance of the findings in this work.
Response 5: To our knowledge, to date no mutations in FANCJ have been identified in the GWAS data for Huntington’s disease (HD), spinocerebellar ataxias (SCAs), or FXD. However, FAN1, a protein partner of FANCJ in a variety of repair pathways, is one of the most important known modifiers of somatic expansion. Both FAN1 and FANCJ interact with MLH1, and both are associated with microsatellite instability. It may be that the two proteins act together to protect against repeat expansion. However, since FAN1’s effect is seen in a broader range of tissues, if indeed FANCJ and FAN1 cooperate in protecting against expansion, other factors must be able to compensate for FANCJ’s loss. We have modified the discussion to mention this.
- Some parts of Figure 1 are duplicated due to typographical errors.
Response 6: This was a problem related to the PDF generated by the journal website. We will try to ensure that it doesn’t happen again.
- “uL” should be corrected to “μL”.
Response 7: Fixed.
Reviewer 2 Report
Comments and Suggestions for Authors
There are some concerns need to address as follows.
1. It would be beneficial for the authors to revise the list of keywords to better reflect the specific scope of the research in this paper.
2. Figure 1 contains overlapping elements, and an extra "A" appears on it. Please correct this figure to ensure the results are presented accurately.
3. Considering the relevance of FXDs to fragile X-associated primary ovarian insufficiency, why the repeat added number in the ovary were not assessed?
Author Response
1. It would be beneficial for the authors to revise the list of keywords to better reflect the specific scope of the research in this paper.
Response 1: Fixed.
2. Figure 1 contains overlapping elements, and an extra "A" appears on it. Please correct this figure to ensure the results are presented accurately.
Response 2: This was a problem related to the PDF generated by the journal website. We will try to ensure that it doesn’t happen again.
3. Considering the relevance of FXDs to fragile X-associated primary ovarian insufficiency, why the repeat added number in the ovary were not assessed?
Response 3: Thanks for the comments, we added the ovary data in the text and as supplementary figure.
Round 2
Reviewer 1 Report
Comments and Suggestions for Authors
I have no additional comments on this manuscript.
Author Response
Reviewer's comments: I have no additional comments on this manuscript.
Response: Thanks.